# A Multi-Channel Packet Scheduling Approach to Improving Video Delivery Performance in Vehicular Networks †

**Pedro Pablo Garrido Abenza *** , **Manuel P. Malumbres** , **Pablo Piñol** and **Otoniel López-Granado**

Department of Computers Engineering, Miguel Hernández University, 03202 Elche, Spain;
mels@umh.es (M.P.M.); pablop@umh.es (P.P.); otoniel@umh.es (O.L.-G.)
* Correspondence: pgarrido@umh.es
† This paper is an extended version of our paper published in Proceedings of the 2022 International Conference on Computational Science and Computational Intelligence (CSCI), Las Vegas, NV, USA, 14–16 December 2022.

**Abstract:** When working with the Wireless Access in Vehicular Environment (WAVE) protocol stack, the multi-channel operation mechanism of the IEEE 1609.4 protocol may impact the overall network performance, especially when using video streaming applications. In general, packets delivered from the application layer during a Control Channel (CCH) time slot have to wait for transmission until the next Service Channel (SCH) time slot arrives. The accumulation of packets at the beginning of the latter time slot may introduce additional delays and higher contention when all the network nodes try, at the same time, to obtain access to the shared channel in order to send the delayed packets as soon as possible. In this work, we have analyzed these performance issues and proposed a new method, which we call SkipCCH, that helps the MAC layer to overcome the high contention produced by the packet transmission bursts at the beginning of every SCH slot. This high contention implies an increase in the number of packet losses, which directly impacts the overall network performance. With our proposal, streaming video in vehicular networks will provide a better quality of reconstructed video at the receiver side under the same network conditions. Furthermore, this method has particularly proven its benefits when working with Quality of Service (QoS) techniques, not only by increasing the received video quality but also because it avoids starvation of the lower-priority traffic.

**Keywords:** VANETs; video delivery; QoS; IEEE 802.11p; multi-channel operation; packet scheduling

## 1. Introduction

Among the applications and services of Intelligent Transportation Systems (ITS), several applications have emerged demanding efficient video-transmission capabilities, from entertainment/consumer-related applications (video conferencing, video surveillance, contextual advertising, tourist information, etc.) and road safety (assisted overtaking, blind spot removal, car insurance support, etc.), to applications that can be crucial for the life of the passengers inside a vehicle (automatic emergency video call—eVideoCall, etc.).

One of the main actors in a video streaming platform is the video encoder. It is the one that provides the compressed video data to be delivered to the network. Its main function is to reduce as much as possible the bitrate required to send one video with the best quality possible. To this end, several standards have emerged during recent decades, with the most popular being the H.264/Advanced Video Coding (AVC) [1], High-Efficiency Video Coding (HEVC) [2], and the recent Versatile Video Coding (VVC) [3] standard. Every new incoming standard halves the average bitrate of a video when it is compared with its predecessor. This evolution is possible at the cost of a significant increase in coding complexity that requires more computational (HW & SW) resources. Every new video coding standard provides new tools that improve the prediction process, increasing both the overall coding complexity and the number of data dependencies required in the encoding

process. So a single error in the received video bitstream may have a higher impact on the decoded video quality since it may affect more pieces of the bitstream due to the higher data dependencies found. This issue makes new video coding standards a bit less error-resilient, in general, with respect to its predecessors.

At this point, when delivering video in error-prone networks, like in Vehicular Ad-hoc Networks (VANETs), we need additional protection schemes to increase the error resilience of the video delivery process as much as possible. So techniques like source coding approaches (e.g., INTRA-refresh, Error Concealment (EC), tile/slice partitioning), channel coding (i.e., Forward Error Correction (FEC) codes), and network level protection (e.g., Quality of Service (QoS), packet interleaving, and Unequal Error Protection (UEP), etc.) should be properly combined to guarantee high-quality video delivery in VANETs.

Focusing on the network point of view, the video transmission over vehicular networks is a challenging task due to the high bandwidth required, the continuously changing network topology (due to the mobility of the network nodes), and the wireless channel's characteristics (shared medium, Doppler effect, signal shading, poor signal coverage, etc.).

When analyzing the network architecture of VANETs, which is driven by the Wireless Access in a Vehicular Environment (WAVE) protocol stack, we find two protocols at the MAC layer: IEEE 802.11p [4] and IEEE 1609.4 [5].

IEEE 802.11p implements Quality of Service (QoS) using the Enhanced Distributed Channel Access (EDCA) mechanism, which allows traffic differentiation of four packet types by means of assigning different packet priorities to each of them. There are four Access Categories (ACs) according to the set of parameters shown in Table 1, such as the Contention Window (CW), the Arbitration Inter-frame Space Number (AIFSN), and the Transmission Opportunity (TXOP). The CW parameter is randomly selected within the range from $CW_{min}$ to $CW_{max}$, which determines the time the node should wait before retrying a transmission (back-off state) when the channel is busy. The AIFSN parameter determines the duration of the Arbitration Inter-frame Space (AIFS), which is the minimum interval time that a node should wait once the wireless channel becomes idle before transmitting a new frame. Finally, the TXOP parameter establishes the time interval during which a node can transmit without contending for access to the wireless channel. IEEE 802.11p introduced a new communication mechanism called Outside the Context of a Basic Service Set (OCB) to support Vehicle-to-Vehicle (V2V) and Vehicle-to-Infrastructure (V2I) communications. According to the IEEE 802.11 standard [6], when the OCB mode is used, the TXOP limit values are set to 0 for each AC. The four access categories, ranging from the one with the highest priority to the one with the lowest priority, are the following: AC_VO (voice), AC_VI (video), AC_BE (best-effort), and AC_BK (background).

**Table 1.** EDCA Parameters of the IEEE 802.11p standard.

| AC | $CW_{min}$ to $CW_{max}$ | AIFSN | TXOP Limit |
|---|---|---|---|
| AC_BK | 15 to 1023 | 9 | 0 ms |
| AC_BE | 15 to 1023 | 6 | 0 ms |
| AC_VI | 7 to 15 | 3 | 0 ms |
| AC_VO | 3 to 7 | 2 | 0 ms |

On the other hand, the IEEE 1609.4 protocol is in charge of multi-channel operation, which works as follows: there is a Control Channel (CCH), through which vehicles transmit safety messages and beacons with vehicle information, and there are four Service Channels (SCHs), through which vehicles transmit the applications' data. The IEEE 1609.4 protocol applies a Time-Division Multiplexing (TDM) to divide the wireless channel into two slots/subchannels, the CCH (control) and the SCH (service), in order to guarantee critical control message exchanges inside a bounded delay window of up to 50 ms. As depicted in Figure 1, at the beginning of each time slot, there is a guard interval (4 ms), which is used to guarantee that every device has completed the switching to the corresponding channel.

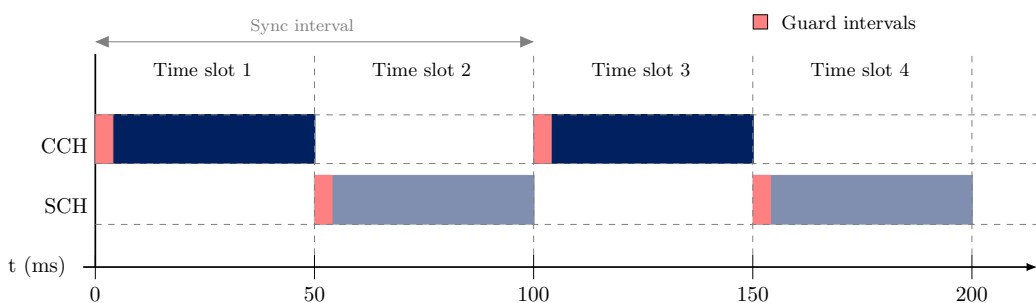

**Figure 1.** Multi-channel operation at the MAC layer for the WAVE architecture.

On the other hand, QoS is highly indicated for video streaming, especially for critical and road-safety-related ITS applications. The use of the AC_VI queue for the transmission of critical video data packets has proven to be very useful in prioritizing video packets over the rest of the network traffic. In a previous work [7], we studied video transmission over VANETs, applying QoS techniques to protect the video stream data packets. Contrary to what was expected, we detected that, in some situations, higher packet losses occurred when assigning a higher priority (AC_VI) to video packets compared with those scenarios where lower priorities were used (AC_BE or AC_BK).

By analysing the experimental results, we noticed that higher packet losses were located just at the beginning of the SCH time slot, and the cause of this issue was the synchronization effect caused by the channel hopping of the IEEE 1609.4 protocol. This happens because the video packets delivered to the MAC layer during the CCH time slot are scheduled for transmission at the beginning of the next SCH time slot. So when SCH time slot starts, we have several delayed packets that need to be immediately sent, increasing the collision probability with traffic sent by other nodes. As a consequence, in most video delivery scenarios, there is a significantly higher rate of packet loss. As it is well known, the loss of several consecutive video packets has a negative effect on the final reconstructed video quality due to the characteristics of the encoded bitstream, reducing the video error resilience (i.e., at the same packet loss ratio, isolated packet losses are less harmful than burst packet losses).

To avoid the undesired synchronization effect described above, we developed a new mechanism, named SkipCCH, aiming to reduce the contention at the beginning of each SCH slot. As a consequence, the packet loss rate is significantly reduced, providing a higher reconstructed video quality. We have implemented SkipCCH at the MAC level to perform a set of simulations to analyze and quantify its benefits when delivering HEVC-encoded video in VANET scenarios.

In this work, we verify the impact of using the proposed mechanism, SkipCCH, whether we use QoS or not. The research questions are the following:

(1) How does activating QoS affect the behavior of video packets?
(2) When we activate QoS, does the proposed mechanism SkipCCH increase the performance of streaming video?
(3) What impact introduces the use of QoS and the SkipCCH mechanism on background traffic?

The remainder of this paper is organized as follows: first, in Section 2, some related works in the literature are presented. Next, in Section 3, the proposed mechanism is represented. In Section 4, the experimental setup is presented, and we define the characteristics of the encoded video sequences (length, bitrate, resolution, etc.), network scenario (area size, nodes, mobility, topology, etc.), simulation parameters (duration, propagation models, background traffic, etc.), and performance metrics (PLR, PDR, PSNR, etc.). The obtained results of the simulation experiments are presented and discussed in Section 5. Finally, in Section 6, conclusions are drawn, and some future work is introduced.

## 2. Related Work

As previously mentioned, Intelligent Transportation Systems open the door to a multitude of new applications and benefits in the field of transportation, which could not have been imagined several years ago. Every day, the number of milestones achieved in this area increases, as well as the visibility and availability of the technologies, which make them possible. Connected vehicle, Smart City (with its related infrastructure), autonomous driving, IoV (Internet of Vehicles), and many others are terms that are acquiring a growing presence in our vocabulary, and all these advances are beginning to play a vital role in our lives, as has progressively occurred with smartphones. Aspects such as road safety [8] (which is one of the main objectives of ITS) both for vehicles and pedestrians, ecological goals [9] (e.g., the reduction in fuel and electricity consumption by optimizing routes), economic purposes [10] (e.g., optimizing and managing transportation fleets), surveillance [11], infotainment [12], and many other targets are achieved by means of ITS. The growing interest in ITS has correspondingly increased the research effort and attention on VANETs, which make ITS possible. Inside this wide field, there are many open lines of research, such as those regarding security [13] (data privacy and integrity, impersonation attacks, vehicle authentication, etc.), protocol standardization [14] (to allow interoperability between vehicles of different brands and with different infrastructures), technical challenges [15] (caused by the mobility of vehicles, dynamic topology, wireless channel, nodes density, etc.), and many others.

An important (with a wide variety of possible applications), as well as challenging, line of research deals with multimedia (audio, image, video) transmission over VANETs. The main characteristics of VANETs (which are error-prone networks), together with the features of multimedia transmission (with high requirements such as timely delivery, high bandwidth needs, data dependency, etc.), make it a challenging task. Many papers in the literature have addressed this line of research. In [16], the authors present a comprehensive review of the main challenges and opportunities for multimedia transmission over VANETs. They highlight the strengths and limitations of transmitting video and images in these networks. Also, they discuss the challenges which arise and the need for Quality of Service in this type of communications. Lastly, they present the opportunities that the transmission of multimedia data may provide; for example, sending short video clips in emergency situations may help rescue services to be more efficient and probably improve their performance in critical situations. In [17], the authors present a literature review of ITS and VANETs. They explain the VANET characteristics and the existing standards, as well as the present challenges and future perspectives on this research area. In [18], Zhou compares the two main different technologies for vehicular networks, i.e., Dedicated Short-Range Communications (DSRCs) and cellular systems, such as Long-Term Evolution (LTE). In both [19,20], the authors focus their research efforts on providing QoS in video streaming over VANETs and, specifically,on how to deliver and maintain it. In [19], specifically, the authors study and compare the existing methods in the literature for achieving QoS and also QoE (Quality of Experience), and they present several QoS and QoE metrics. We use some of these metrics in the present work in order to assess the improvements achieved by our proposals. In [20], the authors propose an architecture for the management of QoS and QoE, based on multilayer video-encoding techniques. In [21], the authors present a literature review centered on the IEEE 802.11p standard and the DSRC technology. They present the features of Medium Access Control (MAC) and Physical (PHY) layers. Our work is aimed at solving some of the problems that appear in these two layers when transmitting video bitstreams over VANETs due to the multi-channel operation of the IEEE 1609.4 protocol.

The issue of the synchronization effect that we have described in the previous section has been studied in works like [22], but not for video streaming applications (with high bandwidth demands and the intrinsic characteristics of compressed video data). In [23], the authors proposed a method based on dynamically changing the AIFSN network parameters in order to better accommodate the variations in network traffic. The goal is to

reduce collisions in the shared medium in order to improve the network performance of audio and video traffic flows. This work was focused on the IEEE 802.11e [24] standard. In [25], the IEEE 802.11p with the WAVE protocol stack was analyzed. One of the main conclusions of this work was that the delay of higher-priority packets is longer than the delay of lower-priority ones. The authors performed analytical and simulated studies of the collision probability of different priority packet flows, showing that high-priority packet flows (i.e., AC_VO) suffer from a higher probability of collisions with respect to low-priority packet flows (i.e., AC_BK), mainly due to their smaller contention window size (Table 1). As a consequence, they observed a higher packet loss rate, which reduces network performance, especially in video streaming applications. Solutions for this issue have been presented in works like [26,27]. In these works, the authors provide a solution, named Video transmission over VANET (VoV), by aligning this transmission to the SCH time slot, by means of a combination of broadcast suppression, store–carry-forward, and rate control mechanisms that order access to the shared channel over one-hop neighbor nodes. Although these works deal with the same problem, our approach (a) is much more simple, as it does not need to exchange messages with the neighborhood (saving network resources); (b) is very easy to implement at MAC layer; and (c) works even better when QoS is enabled. Also, we have explored the use of QoS and we have taken into account the video streaming characteristics (frame rate, MTU, etc.).

## 3. Proposed Method

In this section, we describe the proposed SkipCCH mechanism, which tries to mitigate the problem stated above. This mechanism consists of rescheduling the packet sending time. For this rescheduling, we take into account the channel switching at the MAC layer and also some video streaming constraints (e.g., frame rate and MTU), in order to distribute the packets only during the SCH time slots.

In the IEEE 1609.4 standard (Default), when a video packet arrives at the MAC layer during a CCH time slot, it is scheduled to be sent at the beginning of the next SCH time slot. In order to model the delivery of a video traffic flow, we need to define the packet inter-arrival time ($\Delta$), which is obtained by dividing the total video sequence time by the number of video packets.

If $t_0$ is the time when video streaming begins, then the first packet will be scheduled to be sent at that moment (i.e., $t_1 = t_0$), and the following packets will be scheduled to be sent with an increment of $\Delta$ (i.e., $t_2 = t_1 + \Delta$, $t_3 = t_2 + \Delta$, ..., in general, $t_i = t_{i-1} + \Delta$). The graphical representation of this theoretical scheduling is sketched in the lower part of Figure 2, labeled as "APP".

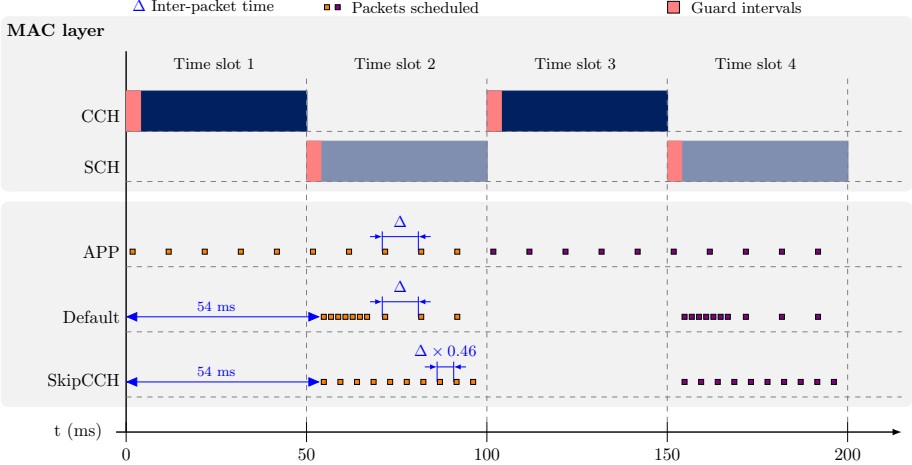

**Figure 2.** Multi-channel operation at the MAC layer for the WAVE architecture (**top**) and packet scheduling for the application (APP), the IEEE 1609.4 standard (Default), and the proposed method (SkipCCH) (**bottom**).

In this operation mode, if a packet is sent at a time $t_i$ that corresponds to a SCH time slot (slots 2, 4, 6, 8, ...), then the MAC layer will try to send it immediately (without extra delays). But if $t_i$ corresponds to a CCH time slot (slots 1, 3, 5, 7, ...), then the packet will remain in the corresponding SCH queue until the next SCH slot arrives. All the queued data packets will be sent as a burst at the beginning of the next SCH slot, as can be seen in Figure 2, leading to a great number of collisions and, consequently, packet losses. The corresponding implementation is depicted in the flowchart in Figure 3, following the red branch "SkipCCH? > Off".

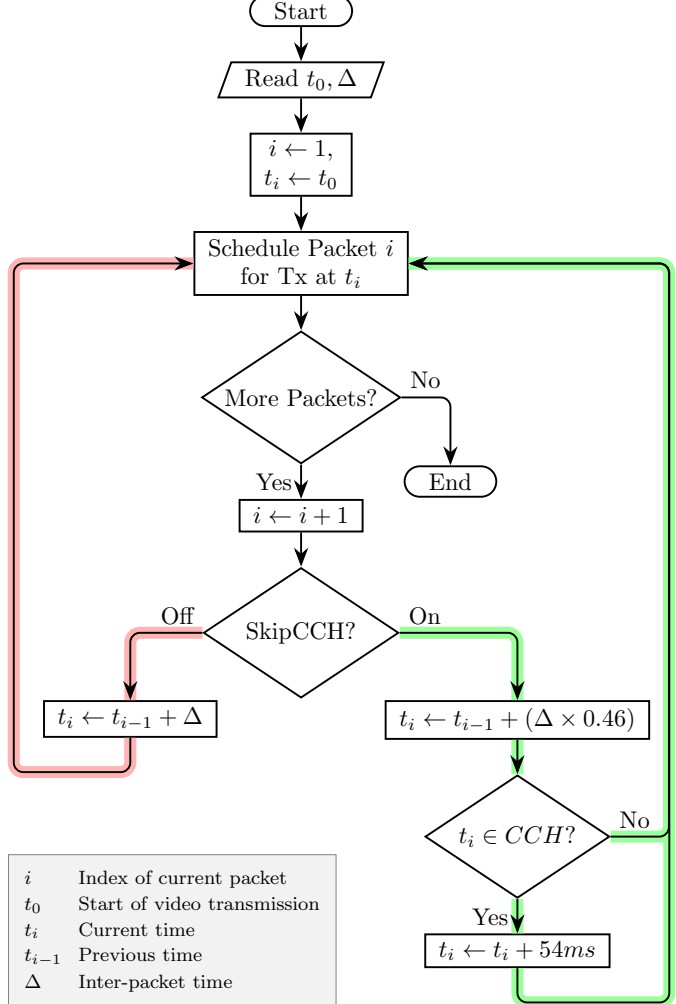

**Figure 3.** Flowchart for the default (SkipCCH Off) and the proposed (SkipCCH On) methods.

To avoid the aforementioned issue, we have proposed a new method (which we have named SkipCCH) [28]. In our proposal, video packets are sent only during SCH time slots. Now, only half of the time is available to send the same number of packets, so, in order to maintain the same sending rate of frames per second, the inter-packet time must be halved ($\Delta \times 0.50$); that is, the delivery frequency has to be doubled. Also, if $t_i$ corresponds to a CCH time slot, then it should be increased by the length of one time slot (50 ms) so that the packet is scheduled within the next SCH time slot. However, in order to be more accurate, we need to take into account the guard intervals (4 ms), which are not used for the transmission of packets. Therefore, the effective length of the time slot is 46 ms (instead of 50 ms). So we have to change the interval between packets to $\Delta \times 0.46$ (a bit shorter) and the time shift needed to avoid the CCH slots to 54 ms (instead of 50 ms). This mode is graphically represented in the bottom part of Figure 2 labeled as "SkipCCH" and

its implementation is depicted in the flowchart of Figure 3, following the green branch "SkipCCH? > On".

*Impact of SkipCCH*

Before analyzing the results of the simulations, the impact of the SkipCCH mechanism on the synchronization effect was studied in detail. For this task, the worst possible case was considered, which corresponds to the highest network load. The time interval considered lasts 0.1 s, that is, one "sync interval" with a length of 100 ms, specifically, the one comprising the CCH slot, ranging from 60.00 s to 60.05 s, together with the next SCH slot, ranging from 60.05 s to 60.10 s.

In Figure 4, we present the default operation (from the application layer point of view), when sending network packets. We have one video flow sent by a Road-Side Unit (RSU) and 10 packet flows delivering background traffic by 10 moving vehicles (`rsu[0]` and `node[1..10]`, respectively). Video data are sent with the AC_VI priority, and background traffic is sent with AC_BK priority. Each background traffic vehicle transmits 100 packets per second (pps). Each square block in the figure indicates the exact time at which each packet is scheduled to be sent to the MAC layer; the size (width) of each block does not mean the duration. As indicated in the legend, video packets are shown in red, and background packets are shown in yellow. As it can be seen, the application layer does not take into consideration whether a data packet (either video or background traffic) is sent during a CCH time slot or not.

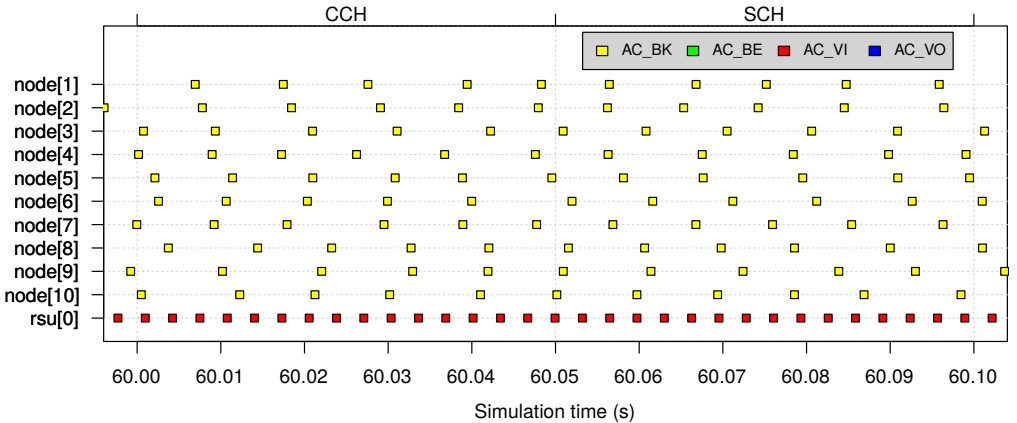

**Figure 4.** Packets scheduled at the APP level for normal operation.

In Figure 5, the MAC scheduling of these packets is shown during the same "sync interval". As shown in Figure 4, each square block indicates the exact simulation time in which each packet is sent to the wireless channel, but here, the size (width) of each block represents its length or duration. Remember that each channel time slot begins with a 4 ms guard interval (which cannot be used for any kind of transmission), as shown in Figures 1 and 2.

As can be seen in Figure 5a, with the default mechanism (SkipCCH off), there is a burst of delayed packets (both video and background traffic packets) that compete for the channel (magnified for clarity), just at the beginning of the SCH time slot. So the involved nodes enter into the back-off state a lot of times, and, as video packets are prioritized by QoS, they win most of the contentions to the detriment of background traffic. This situation leads to a high number of collisions and, as a consequence, to an average increase in video packet delay and, eventually, to packet losses due to the fact of reaching the maximum number of back-off retries.

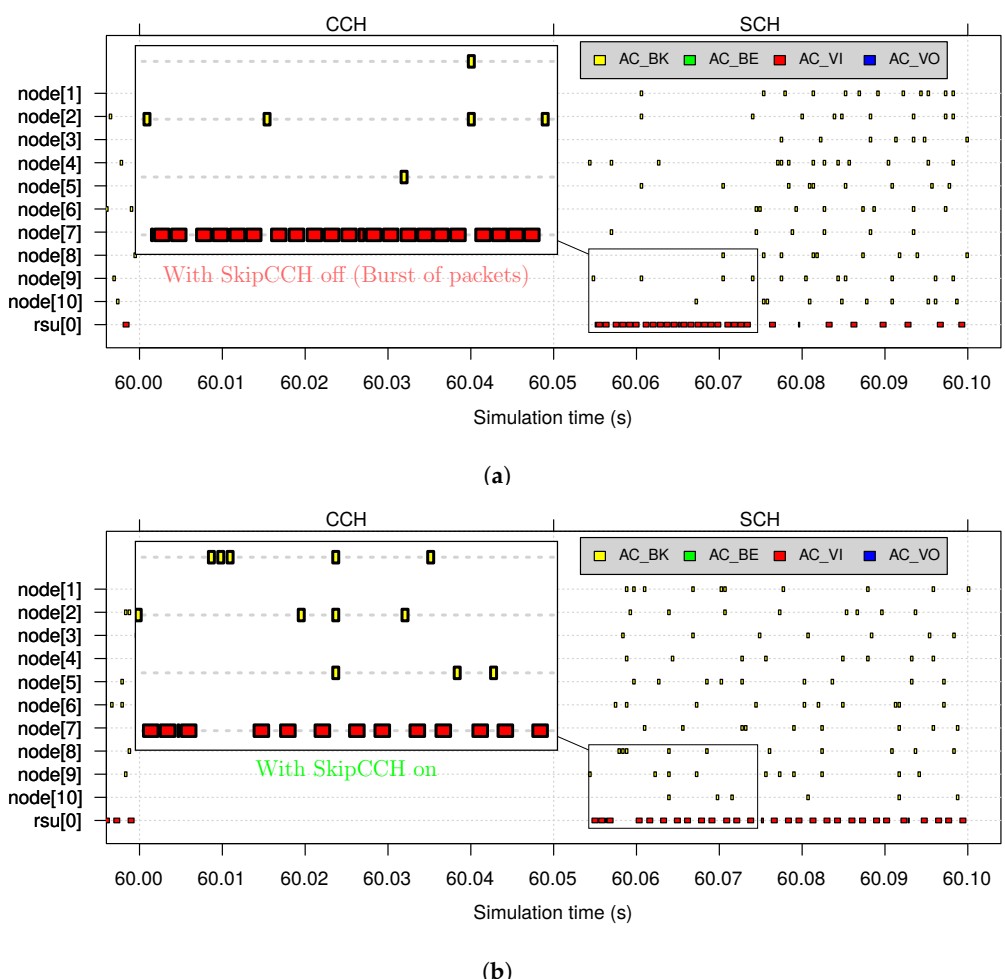

**Figure 5.** Packets sent at MAC level: (**a**) default method, (**b**) proposed method (SkipCCH).

In Figure 5b, we show how data packets are managed at the MAC layer when we apply our SkipCCH proposal (SkipCCH on), which successfully avoids the high-contention phase at the beginning of the SCH time slot. If we focus on the zoomed area, with the default mechanism, we can observe that 22 video packets and very few background packets (6) have been sent. After the burst (t $\simeq$ 60.075 s), the video packets are transmitted at $\Delta$ intervals, and most of the background packets can be sent. On the other hand, with SkipCCH activated, the video packets are uniformly distributed along the SCH time slot, and the zoomed area shows only 14 of them. So the background packets also have a greater probability of being sent, leading to a lower number of collisions. This was the expected result and matches with the theoretical sketch shown in Figure 2. As we will show in the next sections, this behavior brings significant benefits, not only for video traffic but also for low-priority traffic flows.

## 4. Experimental Setup

To evaluate our proposal, we conducted a set of simulations within an urban scenario. We used the Video Delivery Simulation Framework over Vehicular Networks (VDSF-VN) [29], a simulation tool that simplifies the video-encoding and the performance evaluation processes from the design of the appropriate vehicular network scenario (maps, nodes, mobility, etc.) and the simulation management (scheduling, running, results storage) to the processing of evaluation results, via the arrangement of a set of performance metrics and their representation in the form of diagrams, graphs, plots, and reports.

All of these tasks were conducted on a desktop PC equipped with an Intel Core I7-7700K 4.2 GHz quad-core processor with 32 GB DDR4 of RAM running Linux Mint 19.3.

However, all the above-mentioned tools are available for other platforms, such as MacOS X and Windows operating systems. Each of these tasks and the specific software used are further detailed in the next sections.

### 4.1. Video Encoding

For encoding video, we used High-Efficiency Video Coding (HEVC) reference software encoder HM v9.0 [30]. The selected video sequence is named "BasketBallDrill" and belongs to the collection of video sequences included in the Common Test Conditions of HEVC. This video sequence has a resolution of 832 × 480 pixels (Class C) with a bit depth of eight bits, and it is broadcasted in cyclic mode (Figure 6a). It is 500 frames long at a rate of 50 frames per second (fps) in its original form (10 s), but it was sub-sampled at 25 fps with a length of 250 frames to reduce the required network bandwidth.

The encoding of the video sequence has been carried out for two different encoding modes, All Intra (AI) and Random Access (RA). For each encoding mode, a different value for the Quantization Parameter (QP) has been used to achieve an average Peak Signal-to-Noise Ratio (PSNR) of ≈36 dB (see Table 2).

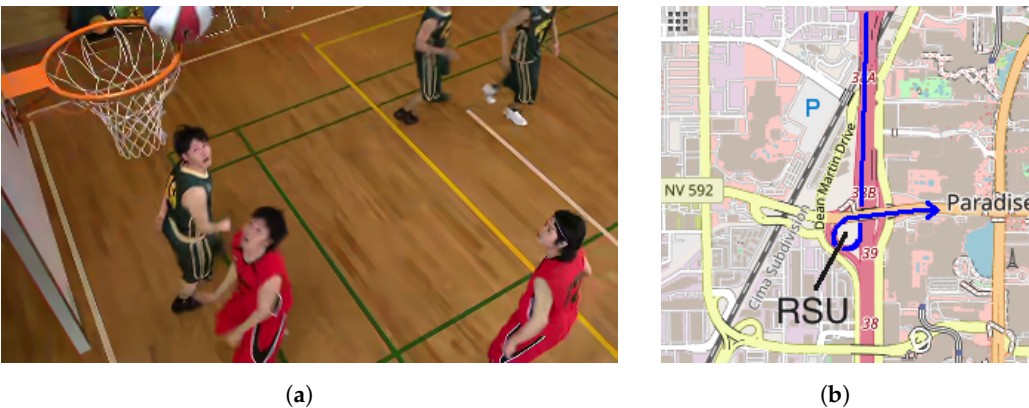

(**a**)                    (**b**)

**Figure 6.** Experimental setup: (**a**) video sequence: BasketBallDrill; (**b**) selected area of Las Vegas.

**Table 2.** Bitstreams generated.

| Mode | QP | Bitrate (Mbps) | PSNR (dB) |
|------|----|----|----|
| AI | 31 | 3.42 | 35.86 |
| RA | 29 | 0.80 | 35.80 |

It can be noticed that, when using the RA encoding mode, the compression performance is significantly higher. This mode provides the same video quality (after decoding) with a significantly lower bitrate demand (four times lower than AI in this case). However, as mentioned before, video flows encoded with the RA encoding mode are more sensitive to packet losses due to the way in which the video is encoded. In this mode, most of the video frames are encoded using the information of other adjacent frames as a reference. So if a packet of a particular frame is lost, this fact harms not only its own frame, but also all the frames that have used it as a reference. One isolated packet loss may impact the reconstruction quality of several frames, multiplying its harmful effect. However, when using the AI encoding mode, all frames are encoded without reference to other frames (there are no frame dependencies), so one isolated packet loss will only affect the frame to which it belongs (no error propagation is produced). The decision to use one encoding mode or the other should be based on the trade-off of available bandwidth and video error resiliency. To evaluate the SkipCCH proposal, we have used two different setups. One of them uses the AI video encoding mode, which is able to reach network saturation levels when high background traffic levels are used, and the other setup uses the RA video encoding mode with moderate network traffic loads.

### 4.2. Network Scenario and Simulation Parameters

The network scenario (Figure 6b) consists of a real map of a 2000 m × 2000 m square area of Las Vegas (around Dean Martin Drive), downloaded from OSM (Open Street Map) database [31], and converted to a format that can be handled by the simulators. As mentioned before, we have used the VDSF-VN simulation framework, which is composed of public domain simulators like the OMNeT++ v5.6.2 [32] together with the Veins (VEhicles In Network Simulation) v5.1 framework [33], and the SUMO (Simulation of Urban MObility) v1.8.0 as the mobility simulator [34]. In addition, a set of third-party libraries (OSM, R, gnuplot, HEVC, etc.) was also used.

In Table 3, we summarize some parameters that were defined for all the simulation tests and are described just below. One Road-Side Unit (RSU) was placed near the center of the scenario (`rsu[0]`). This RSU is a static node that is in charge of broadcasting the source video sequence. Also, there are 11 vehicles that drive near the RSU (the route followed is represented by a blue arrow in the scenario). One of them is the video client (`node[0]`), the one that receives the video stream. The other 10 vehicles (`node[1..10]`) pursue the video client vehicle (following just the same route) injecting background traffic at different rates to simulate different network loads, ranging from no background traffic to network loads that lead to channel saturation. Each of the 10 vehicles injects packets with a payload of 512 bytes (4096 bits) at different packet rates (0, 12, 25, 50, 75, and 100 pps), providing a total background traffic of 0, 0.49, 1, 2, 3, and 4 Mbps, respectively. In addition, more vehicles were inserted (`node[11..60]`), which just drive around the scenario without generating traffic and following randomly generated routes. All the vehicles move at a maximum speed of 14 m/s (50 km/h) and the total time of the simulation is 200 s.

**Table 3.** Simulation parameters.

| Parameter | Value |
| --- | --- |
| City | Las Vegas |
| Simulation area | 2000 m × 2000 m |
| Simulation time | 200 s |
| Number of RSUs | 1 (`rsu[0]`) |
| Number of client vehicles | 1 (`node[0]`) |
| Number of background vehicles | 10 (`node[1..10]`) |
| Background traffic load | {0, 12, 25, 50, 75, 100} pps |
| Max. speed of the vehicles | 14 m/s (50 km/h) |

The communication range for the RSU, as well as for all the vehicles, is around 500 m, which is the default value used in Veins. Also, obstacles, such as the buildings in the city, were modeled by means of the `SimpleObstacleShadowing` propagation model. Some representative parameters of the network interfaces are shown in Table 4.

**Table 4.** PHY/MAC parameters.

| Parameter | Value |
| --- | --- |
| Carrier frequency | 5.890 GHz |
| Propagation model: | SimpleObstacleShadowing |
| Bitrate | 18 Mbps |
| Transmit power | 20 mW |
| RX Sensitivity | −89 dBm |
| Communication range | 510.87 m |
| MAC queue size | 0 (infinite) |

*4.3. Performance Metrics*

During the simulations, we used several performance metrics that may be grouped into two sets: network and application performance metrics. The metrics for the network performance are as follows:

- Packet Delivery Rate (PDR) measures the ratio between the packets that were correctly delivered and the total number of packets sent by one particular node. It use to be represented as a percentage value (%).
- Application Throughput (aka Goodput) measures the number of bits per second that were delivered to the receiver. It is similar to the PDR but with an absolute value and takes into account the size of packets. It is expressed in bits per second (bps).
- Average End-to-End Delay is a measure that shows the average delay of sent packets from the application point of view. In other words, it represents the elapsed time from the instant in which the source application sends one packet until the instant in which it is received by the destination application. It is represented in milliseconds (ms).
- Jitter measures the average variance of packet delays. This measure gives us an idea about the stability of the communications.
- Packet Lost Rate (PLR) is a measure that shows the channel packet error rate. It is expressed as the ratio between the number of lost packets and the total number of packets sent. It is represented as a percentage (%).

The other set of performance metrics are related to the application performance. In this case, they are metrics related to video performance, giving us a prediction of the user's Quality of Experience (QoE). The selected performance metrics are the following:

- Frame Lost Rate (FLR) is the ratio between the number of frames lost (unable to be decoded) and the total number of video frames. Although it could give us the same information as PLR, it sometimes does not have the same behavior. For example, if one encoded frame is decomposed in N packets and one of them is lost, this frame would not be decoded at the destination, so it would be counted as a lost frame. This behavior may be different since it depends on the error-resilience mechanisms applied by the video encoder (e.g., tiles/slices frame partitioning). In our case, no error resilience mechanism was applied by the encoder. It is represented as a percentage (%).
- Peak Signal-to-Noise Ratio (PSNR) is a well-known objective assessment video quality metric that represents the quality degradation level of a frame or video (average value of all frames) when affected by noise (compression, bit/packet errors, filtering, etc.). In our case, it will determine the final video quality that the user may perceive from the received video streaming. It is based on the mean squared error (MSE) of frame pixels with respect to the original frame. The MSE is transformed into a logarithmic scale, as can be shown in Equation (1). It is measured in decibels (dB).

$$PSNR(n)_{dB} = 20 \cdot \log_{10}\left(\frac{255}{\sqrt{MSE(n)}}\right),$$

$$MSE(n) = \frac{1}{N_{col} \cdot N_{row}} \sum_{i=1}^{N_{col}} \sum_{j=1}^{N_{row}} [Y_S(n, i, j) - Y_D(n, i, j)]^2$$

(1)

- Structural Similarity Index Measure (SSIM), like PSNR, is a perceptual video quality metric computed frame-by-frame and averaged for all the frames. SSIM analyses similarities between the luminance, contrast, and structure of both the original and reconstructed (decoded) frames [35]. The value of SSIM is typically in the range [0, 1], where 1 indicates no differences. In Equation (2), we show the expression to compute SSIM,

$$SSIM(S, D) = \frac{(2\mu_S\mu_D + C_1) \cdot (2\sigma_{SD} + C_2)}{(\mu_S^2 + \mu_D^2 + C_1) \cdot (\sigma_S^2 + \sigma_D^2 + C_2)}$$

(2)

where $S$ and $D$ represent the source and decoded images, $\mu_S$ and $\mu_D$ are the mean difference for all the pixels of each frame with respect to the mean luminance of the frame, $\sigma_S$ and $\sigma_D$ are the standard deviations used to measure the contrast, $\sigma_{SD}$ is the cross-covariance of the two frames $S$ and $D$, and $C_1$ and $C_2$ are constants to adjust the SSIM final score.

## 5. Results and Discussion

In this section, the simulation results for the two defined setups, that is, for the two video-encoding modes (AI and RA), are shown and discussed. We have run our simulations for the six different background traffic rates mentioned before (from 0 to 100 pps) and enabling and disabling both QoS and SkipCCH. The results obtained from this comparison for both video-encoding modes are presented and discussed below. All the statistics shown correspond to the video sequence received by the client (`node[0]`) between t = 60 s and t = 70 s.

### 5.1. All Intra Mode

Starting with the experiments related to the AI encoding mode, in Figure 7a, we show the PDR under different background traffic rates and, in Figure 7b, the throughput obtained by the application. It can be seen that when the QoS is not used (QoS off), video traffic is greatly affected as the network becomes saturated. On the other hand, when using Quality of Service (QoS on), video traffic has a greater resilience, which is very noticeable for medium and high network loads, at the expense of background traffic, as will be shown later.

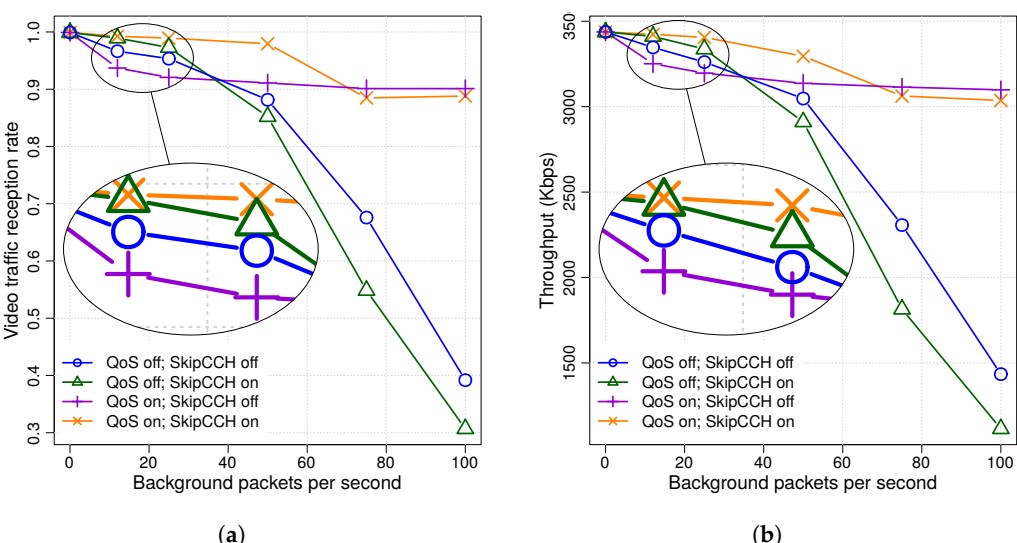

(**a**)  (**b**)

**Figure 7.** AI—reception of video packets (`node[0]`): (**a**) PDR, (**b**) application throughput.

It is worth highlighting a specific case, which was the one that led us to the development of the proposed mechanism. Contrary to what might be expected, when using QoS with the default mechanism (QoS on; SkipCCH off), the video PDR was lower than that when not using QoS (QoS off; SkipCCH off) for low to medium network loads (12 to 25 pps), zoomed in for clarity in Figure 7. The reason for this was the synchronization effect due to the channel switching of the MAC layer, which increases the probability of collision for higher-priority ACs [25]. As can be seen in Figure 7, the SkipCCH mechanism solves this situation since the reception rate of video traffic is higher for low- and medium-network loads, whether QoS is activated (QoS on; SkipCCH on) or not (QoS off; SkipCCH on). However, when using QoS, the improvement achieved by SkipCCH is even higher at these network traffic loads.

Figure 8a shows the average End-to-End delay of the video packets broadcasted by the server (`rsu[0]`). The first thing to notice is that when the QoS is not used (QoS

off), the delay increases heavily when the network is saturated, regardless of whether the SkipCCH mechanism is used or not, while when using QoS (QoS on), the delay remains nearly constant since video packets have a greater transmission opportunity than background traffic.

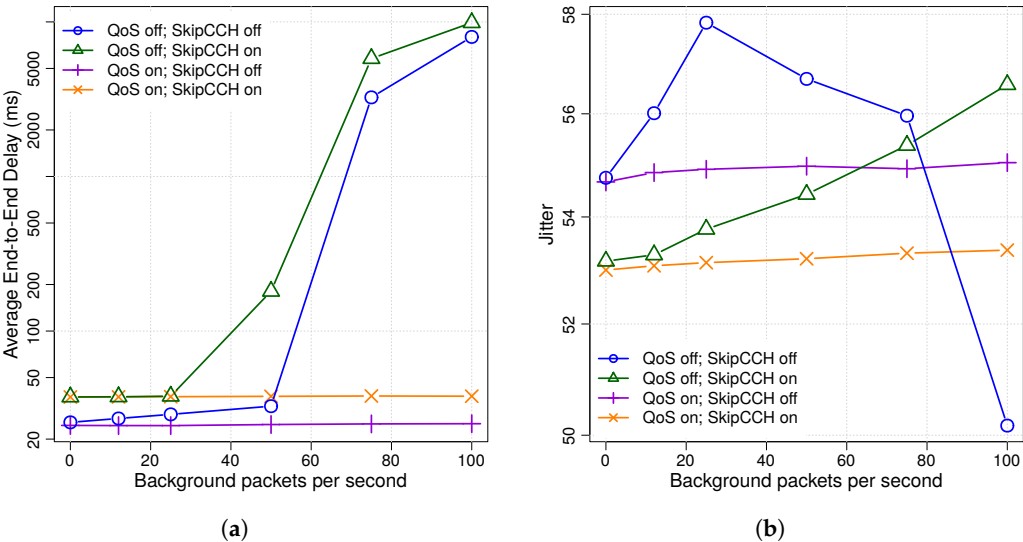

**Figure 8.** AI—(**a**) end-to-end delay. (**b**) Jitter for AI.

With respect to the use of the SkipCCH mechanism, a general conclusion is that it always increases the average end-to-end delay, mainly due to the uniform spreading of delayed video packets in the SCH time slot, which avoids the bursts at the beginning of the slot. The increase in the average video packet delay is around 13 ms when QoS is activated and remains more or less constant at all network traffic loads. At this point, it should be noticed that the 802.11p MAC layer imposes a delay to those packets that arrive during the CCH period slot (up to 54 ms in the worst case), so for critical delay-constrained video applications, these limitations may be highly restrictive. However, this is not an issue for most real-time video streaming applications, whose requirements are not so strict. For example, the delay should not exceed 200 ms for a safe overtaking maneuver nor more than 6 s for video surveillance [36].

Regarding *jitter*, Figure 8b clearly shows lower and constant values (for any traffic load) when QoS is activated. The combination with the SkipCCH proposal activated is the one that obtains better *jitter* values. However, the differences in *jitter* values are not very significant in this case.

Figure 9a shows the video packet loss rate due to transmission errors, and Figure 9b shows the number of frames that cannot be decoded. Remember that the loss of a single packet can cause a video frame to become undecodable. Therefore, both graphs show a similar trend, although the loss of many packets, especially if they follow burst patterns, does not affect the loss of frames as much (i.e., a burst loss of three packets does not imply the loss of just three frames).

As can be seen in these graphs, when QoS is not used, there is a packet loss of 36% and 44% in the worst case, producing a frame loss rate of up to 98%. On the other hand, when QoS is used with the default mechanism (SkipCCH off), the positive effect is appreciated only when the network starts to be saturated (50 or more pps of background traffic). However, contrary to what one might expect, when the network is slightly saturated (12 or 25 pps), both packet loss and frame loss rates are higher. As stated before, this is due to the synchronization effect of the channel switching of the IEEE 1609.4 protocol. At high network loads, the situation is just the opposite, being slightly better when not using SkipCCH. When SkipCCH is enabled, the inter-packet arrival time of video packets is reduced ($\Delta \times 0.46$ instead of $\Delta$) in order to accommodate all video packets (the ones scheduled in CCH and SCH time slots) in regular intervals in the SCH time slot. As previously explained, it helps



to reduce collisions at the beginning of the SCH slot, but at high network loads, the number of video packet collisions across the SCH slot increases in such a way that it surpasses the ones produced when not using SkipCCH.

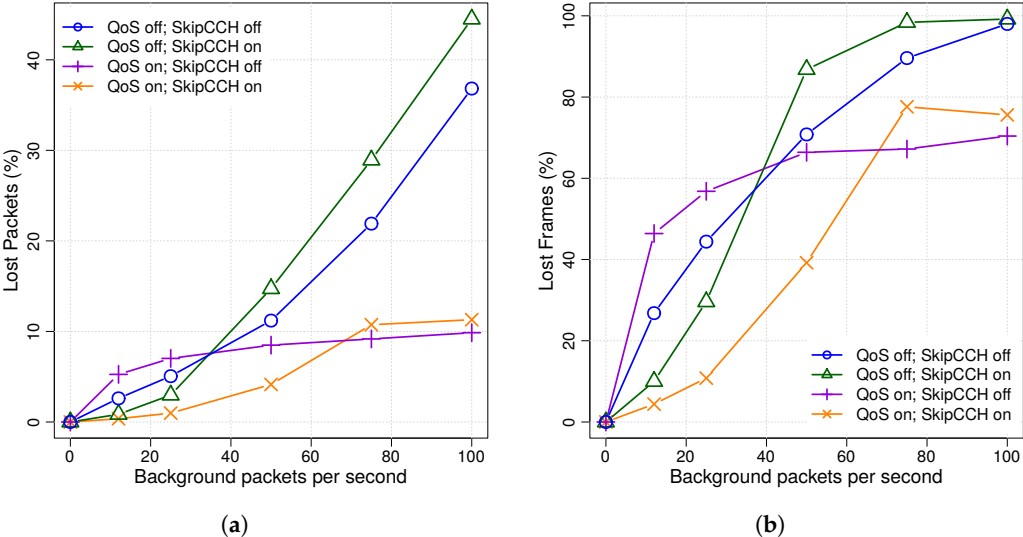

**Figure 9.** AI—(**a**) lost video packets. (**b**) lost frames.

When enabling SkipCCH, it can be seen that both the packet and frame loss rates are significantly reduced. For example, without Quality of Service, the frame loss rate (Figure 9b) at low network loads (12, 25 pps) reaches 26% (SkipCCH on) and 44% (SkipCCH off). When enabling QoS and the SkipCCH method, the frame loss rate drops from 46% to 4% (at 12 pps) and from 56% to 10% (at 25 pps).

Finally, to check the effect of the lost frames on the perceived video quality, two quality metrics were analyzed, the PSNR and the SSIM. Both metrics show a very similar trend (Figure 10), roughly inverse to the lost frames graph (Figure 9b).

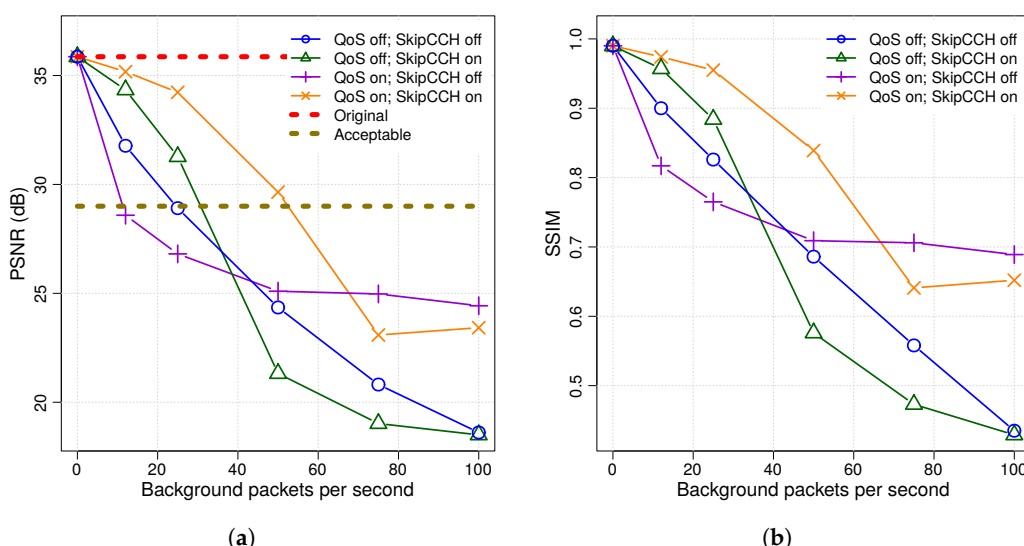

**Figure 10.** AI—video quality measurements: (**a**) PSNR, (**b**) SSIM.

In Figure 10a, PSNR quality measurement, we have defined two thresholds: (a) an upper threshold of 35.86 dB as the quality of the encoded video (*Original* label), and (b) the lower threshold of 29 dB (*Acceptable* label), considered as the minimum video quality level established by the application. The explanation of these results is in accordance with the ones given for the packet/frame loss rates, showing that the best option is to enable both

QoS and SkipCCH mechanisms to achieve at least an acceptable video quality for low to moderate network traffic loads.

In Table 5, we show the video quality results by means of a video quality color table. We define four quality ranges: (a) ≥32 dB Excellent (green), (b) [29..32) dB Acceptable (yellow), (c) [25..29) dB Low (orange), and (d) <25 dB Very Low (red). Considering the red (unusable) and the orange cells (unacceptable), the only configuration that provides excellent or acceptable quality levels for low to moderate network loads is the one that uses QoS in combination with SkipCCH. However, the use of SkipCCH is highly recommendable in any case (using QoS or not).

**Table 5.** AI—PSNR of reconstructed video sequence (dB units).

| | | Background Packets per Second (pps) | | | | | |
|---|---|---|---|---|---|---|---|
| QoS | SkipCCH | 0 | 12 | 25 | 50 | 75 | 100 |
| off | off | 35.86 | 31.78 | 28.91 | 24.35 | 20.81 | 18.61 |
| off | on | 35.86 | 34.34 | 31.28 | 21.32 | 19.02 | 18.50 |
| on | off | 35.86 | 28.59 | 26.81 | 25.10 | 24.97 | 24.43 |
| on | on | 35.86 | 35.18 | 34.23 | 29.65 | 23.09 | 23.42 |

PSNR of the original video sequence: ≈35.86 dB. ≥32 dB Excellent (green); [29.32) dB Acceptable (yellow); [25.29) dB Low (orange); <25 dB Very low (red).

### 5.2. Random Access Mode

In this section, we have repeated the same set of experiments as those in the previous one, but changing the video encoding mode from AI to RA. So we have a setup where a low bandwidth video flow (only 0.8 Mbps) is delivered with a high degree of sensitivity to packet losses. First, we will review the performance of the main network statistics, and then, we will focus on analyzing the impact of packet losses on the final video quality.

In Figure 11, we show the PDR, end-to-end delay, and PLR statistics. All of them show a behavior that is similar to that of the statistics explained in the previous section. It should be taken into account that this network setup never reaches high network loads and is always far from network saturation. This fact explains the differences between both setups in terms of absolute values. In Figure 11a, we can see the PDR curves, where the best option (QoS on; SkipCCH on) achieves values higher than 94% for all background traffic loads. In Figure 11b, the end-to-end delay confirms that, when using SkipCCH, there is an average increase around 16 ms (≈75%). In Figure 11c, the PLR curves are shown. In this setup, the differences between using or not using the SkipCCH method are more relevant, which is especially significant for medium to high background traffic loads. However, when QoS is enabled, the differences are not so relevant, keeping PLR rates from 0.33% to 6% when SkipCCH is also enabled.

Now, we will analyze the error resilience capability of the RA-encoded video stream. In Figure 12, we show the video quality results using both the PSNR and SSIM metrics. We use the same PSNR quality thresholds as the ones employed in the previous section. It can be seen that, if SkipCCH is disabled and packet losses start appearing (at 12 pps), the video quality drops to extremely low quality levels, maintaining this unacceptable quality at the rest of the background traffic loads. This proves that the RA-encoding mode is highly sensitive to packet losses (as explained before). By enabling SkipCCH and QoS, the quality keeps above the lower threshold until there are moderate background traffic loads.

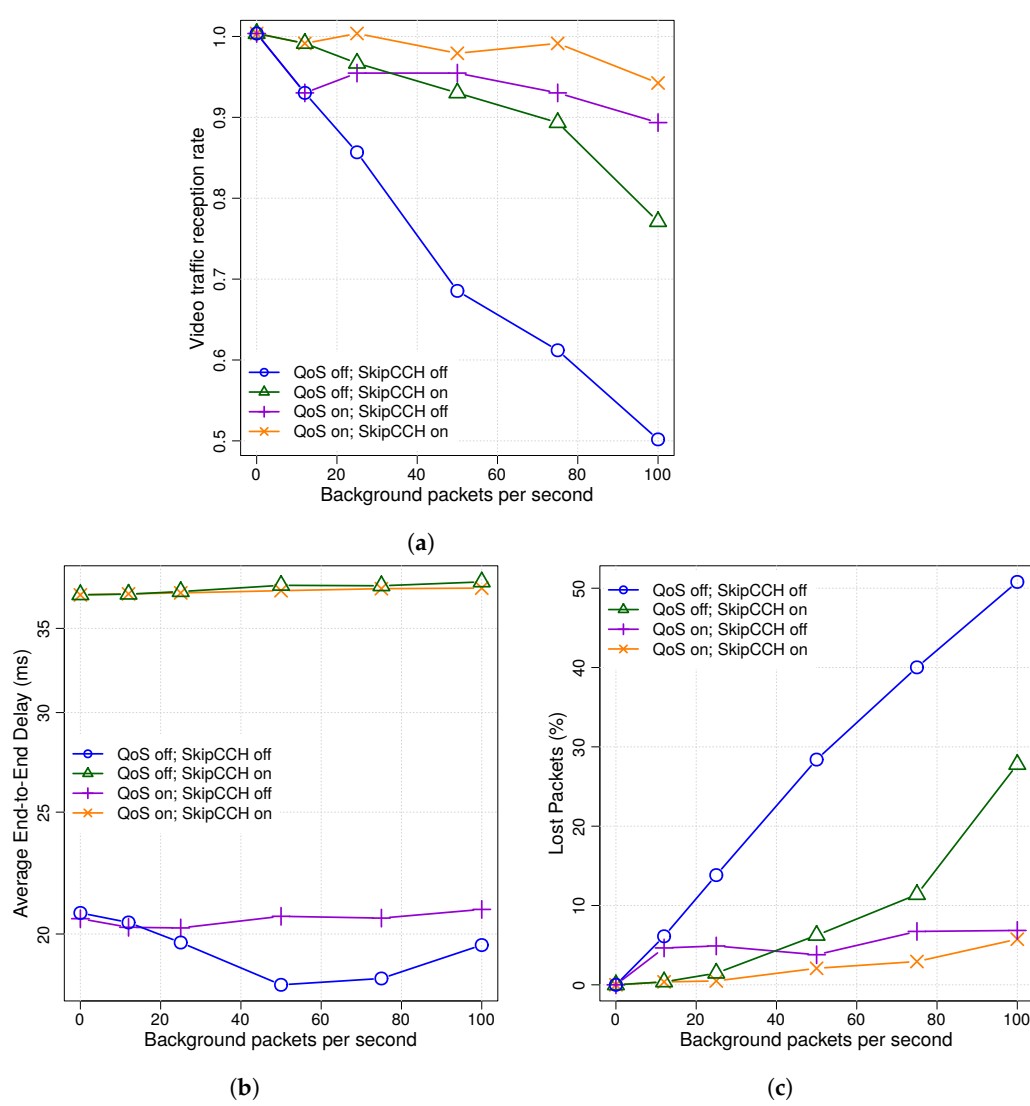

**Figure 11.** RA—network statistics (`node[0]`): (**a**) PDR, (**b**) end-to-end delay, (**c**) PLR.

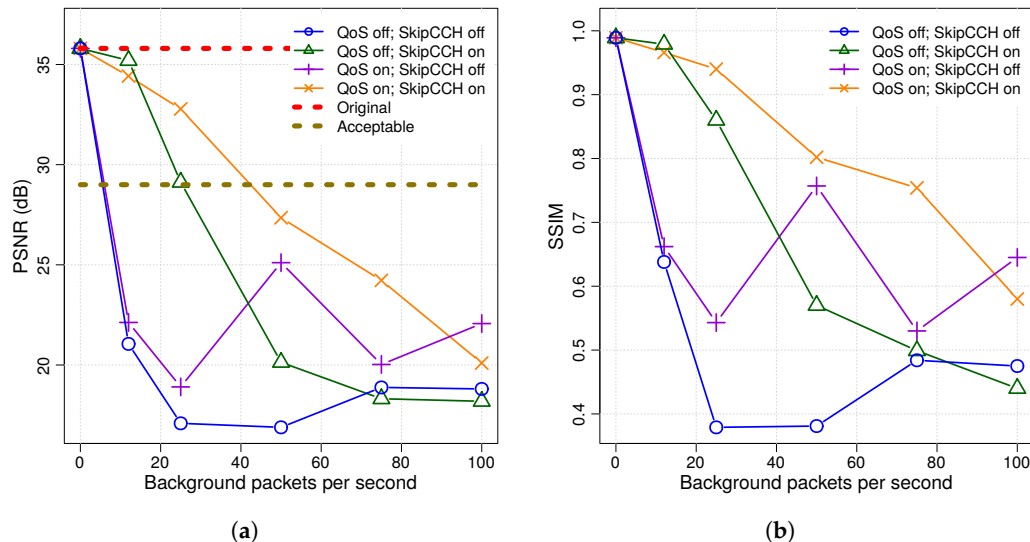

**Figure 12.** RA—video quality metrics: (**a**) PSNR, (**b**) SSIM.

We may notice the pseudo-random behavior of PSNR when the SkipCCH is disabled (especially when QoS is on) at medium to high network loads. This is due to (a) the use

of the HEVC RA video coding mode, which is much more sensitive to packet losses (as explained before) and (b) the effect of using the "frame-copy" error-concealment (EC) method that replaces a lost frame with the last frame correctly received. So when one packet is lost, the frame it belongs to is also lost, being replaced (frame-copy) by the last received frame. Notice that this replacement may affect the final average video quality (PSNR/SSIM) shown in Figure 12. So, depending on where packets are lost, the final average video quality may vary, which is one of the causes of these fluctuations in video quality, especially when SkipCCH is disabled. Also, we have to remark that we are working out of the operative working space (under 25 dB, it is considered a very poor video quality), where difference in PSNR values are not appreciated by the user.

Finally, as we have shown previously, in Table 6, we show a color table representing different quality levels for the RA video encoding mode. At first glance, we notice that, if we disable SkipCCH, the resulting video quality is completely unacceptable for all traffic background loads. Furthermore, by using QoS and SkipCCH, the quality levels keep above the lower quality threshold until moderate background traffic loads. Notice that this setup only reaches moderate network traffic loads (far from network saturation), and if the background traffic increases, the video quality may drop even more.

**Table 6.** RA—PSNR of reconstructed video sequence (dB units).

| QoS | SkipCCH | Background Packets per Second (pps) | | | | | |
|-----|---------|------|------|------|------|------|------|
|     |         | **0** | **12** | **25** | **50** | **75** | **100** |
| off | off | 35.80 | 21.05 | 17.09 | 16.89 | 18.88 | 18.81 |
| off | on  | 35.80 | 35.20 | 29.12 | 20.13 | 18.32 | 18.19 |
| on  | off | 35.80 | 22.12 | 18.91 | 25.11 | 20.03 | 22.07 |
| on  | on  | 35.80 | 34.42 | 32.78 | 27.35 | 24.23 | 20.10 |

PSNR of the original video sequence: ≈35.80 dB. ≥32 dB Excellent (green); [29..32) dB Acceptable (yellow); [25..29) dB Low (orange); <25 dB Very low (red).

### 5.3. Background Traffic

As well as the protection of video packets, another objective to be considered is not to excessively penalize the rest of the network traffic. Figure 13 shows how background traffic is affected under different scenarios by enabling/disabling QoS and SkipCCH for both video-encoding modes.

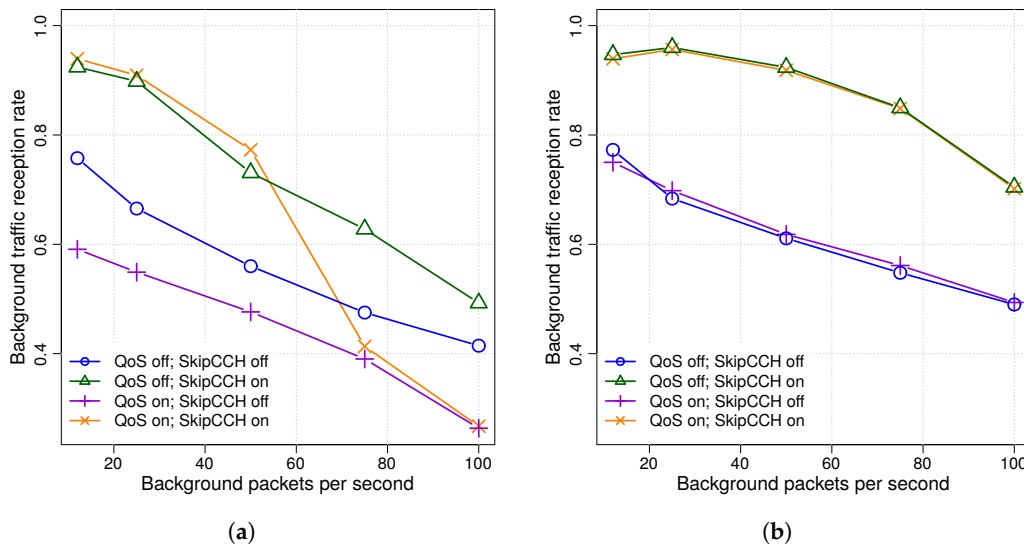

**Figure 13.** Background traffic PDR (`node[0]`): (**a**) AI mode, (**b**) RA mode.

In Figure 13a, we show that just by enabling SkipCCH, the PDR of background traffic greatly improves. In particular, when there is no QoS, the PDR increases by up to 23% (16% on average), and with QoS, the increase is even higher, up to 36% (20% on average). In the last case (QoS enabled), when using SkipCCH, the PDR improvements are negligible at high to saturated network loads, mainly due to the prevalence of priority video traffic. In Figure 13b, we show the same scenario but using the RA video encoding mode, which requires much less bandwidth than the former one, so the network load is always far from saturation. Here, we may better appreciate the benefits of our SkipCCH proposal, either using QoS or not.

Therefore, we can come to the conclusion that applying the SkipCCH mechanism to the video transmissions reduces the negative impact of using QoS on the rest of the network traffic; that is, a better use of the wireless channel is achieved.

## 6. Conclusions

In this work, a method named SkipCCH was proposed to improve the performance of video streaming applications over VANETs. The proposed mechanism skips the CCH time slots in the scheduling of video packets and properly distributes this scheduling within the next SCH time slot, reducing the channel contention and, as a consequence, the packet loss ratio. It has shown good performance in all conditions, especially when it is combined with QoS. In addition, SkipCCH is able to reduce the typical starvation effect that QoS imposes to lower priority traffic, allowing more resources (transmission opportunities) for this kind of traffic while keeping the QoS prioritization schema.

In future work, we plan to improve the SkipCCH method by analyzing different scheduling strategies using the available information in each node. The goal will be to improve network performance by reducing the network contention as much as possible. These strategies must take into account the amount of end-to-end network delay that they introduce, in order to find a trade-off between reducing network contention and its impact over the average end-to-end delay of packets.

On the other hand, the reconstructed video quality may be improved when working from moderate to high network traffic loads, especially for those videos encoded with the RA configuration. We will also search for additional mechanisms to protect, even more, the transmission of encoded video flows in VANETs (source and channel coding, UEP, etc.) in order to provide the transmission o high video quality in a wide range of network conditions (network contention, channel interference, propagation errors, etc.).

**Author Contributions:** Funding acquisition, O.L.-G.; Investigation, P.P.G.A., M.P.M. and P.P.; Software, P.P.G.A. and P.P.; Supervision, M.P.M. and P.P.; Validation, M.P.M. and P.P.; Writing—original draft, P.P.G.A., M.P.M., P.P and O.L.-G.; Writing—review and editing, P.P.G.A., M.P.M., P.P. and O.L.-G. All authors have read and agreed to the published version of the manuscript.

**Funding:** This work is part of the grant PID2021-123627OB-C55, funded by MCIN/ AEI/ 10.13039/ 501100011033 and by "ERDF A way of making Europe" and from the Valencian Ministry of Innovation, Universities, Science and Digital Society (Generalitat Valenciana) under Grant CIAICO/2021/278.

**Data Availability Statement:** The software used in this work is available from the SourceForge repository at https://sourceforge.net/projects/vdsf-vn-ppp-qos/ (accessed on 18 October 2023.)

**Conflicts of Interest:** The authors declare no conflicts of interest.

## Abbreviations

The following abbreviations are used in this manuscript:

| | |
|---|---|
| AC | Access Category |
| AIFSN | Arbitration Inter-frame Space Number |
| AVC | Advanced Video Coding |
| CCH | Control Channel |
| CW | Contention Window |

| EDCA | Enhanced Distributed Channel Access |
|------|-------------------------------------|
| FEC | Forward Error Correction |
| FLR | Frame Loss Rate |
| HEVC | High-Efficiency Video Coding |
| ITS | Intelligent Transportation System |
| MAC | Medium Access Control |
| MTU | Maximum Transmission Unit |
| OCB | Outside the Context of a Basic Service Set |
| PDR | Packet Delivery Rate |
| PLR | Packet Loss Rate |
| PSNR | Peak Signal-to-Noise Ratio |
| QoE | Quality of Experience |
| QoS | Quality of Service |
| QP | Quantization Parameter |
| RSU | Road-Side Unit |
| SCH | Service Channel |
| SSIM | Structural Similarity Index Measure |
| TXOP | Transmission Opportunity |
| V2I | Vehicle-to-Infrastructure |
| V2V | Vehicle-to-Vehicle |
| VANET | Vehicular Ad hoc Network |
| VVC | Versatile Video Coding |
| WAVE | Wireless Access in Vehicular Environments |

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
