# Peer review of "A Multi-Channel Packet Scheduling Approach to Improving Video Delivery Performance in Vehicular Networks†"

_computers, doi:10.3390/computers13010016_

Round 1

Reviewer 1 Report

Comments and Suggestions for Authors

This article proposes a method to improve the received quality of video content transmitted over vehicular networks. The subject is interesting because several applications have emerged for ITS that involve the use of video. However, the paper suffers from several limitations. The positioning in relation to the state of the art is weak, the bibliographic references are old and many recent works are not mentioned. Furthermore, the scientific contribution is not significant (barely one page including a half-page diagram out of the 16 pages in total) and it is relatively limited. Moreover, the problem is not put into equation and the parameters involved are poorly described. Here are the main comments:

1.      the title is too generic and gives the impression that the paper is a survey when the paper presents a specific and limited contribution. Likewise, the list of keywords is imprecise and not adapted.

2.      there is a severe lack of references and more particularly recent references in the article, while the subject of the research gives rise to numerous publications, for example:

M. Okpok, B. Kihei, “Challenges and Opportunities for Multimedia Transmission in Vehicular Ad Hoc Networks: A Comprehensive Review”, Electronics 2023, 12(20), 4310

B. Alaya et al., “Multilayer Video Encoding for QoS Managing of Video Streaming in VANET Environment”, ACM Transactions on Multimedia Computing, Communications, and Applications, 18(3), No. 82, pp 1–19, 2022

B. Alaya et al., “Study on QoS Management for Video Streaming in Vehicular Ad Hoc Network (VANET)”, Wireless Personal Communications volume 118, pages 2175–2207 (2021)

M-A. Labiod et al., “Enhanced adaptive cross-layer scheme for low latency HEVC streaming over Vehicular Ad-hoc Networks (VANETs)” , Vehicular Communications, Volume 15, 2019, Pages 28-39

Al-shareeda, M.A. et al., “A Comprehensive Survey on Vehicular Ad Hoc Networks (VANETs). In Proc. of the 2021 International Conference on Advanced Computer Applications (ACA), Maysan, Iraq, 25–26, July 2021, 156–160.

3.      In Table 1, the parameters CWmin-max, AIFSN, TXOP limit are not defined.

4.      Overall, the structure of the paper is not clear and makes the paper difficult to understand for the reader. For example, Figure 1 already mentions the proposed SkipCCH method before it has been introduced, and the delta parameter used several times in the figure is not known or explained.

5.      The authors should consider VVC instead of HEVC, because VVC is the new state-of-the-art video coding standard.

6.      Figure 3 is not useful (not informative).

7.      Figure 5 a) and b) are too small and difficult to analyze.

8.      The experimental part is a bit verbose and some analyzes are missing in Section 5. In Figure 8, how do the authors explain that the QoS on/SkipCCH on solution is less good than the QoS on/SkipCCH off one for high background packets per second values? In Figure 11, how can we explain the quasi-random behavior of the PSNR and the SSIM for the blue, purple and green curves? One expects monotonically decreasing curves, how can the quality increase when the number of background packets per second increases?

Comments on the Quality of English Language

No specific comments on English language. The text is sometimes verbose and could be reduced, for example in Section 4.

Reviewer 2 Report

Comments and Suggestions for Authors

This paper proposed a SkipCCH mechanism to improve video delivery performance in vehicular networks. Simulations are carried out to evaluate the performance of the proposed method in terms of several metrics, including traffic reception rate, throughput, and delay. I have several major concerns in the paper, which are listed as follows:

1. The dynamic analysis of network topology should be more reflected in the article

2. The authors claimed that "With our proposal, streaming video in vehicular networks will provide better reconstructed video quality at the receiver side under the same network conditions", however, the current version of the paper fails to show this in simulations.

3. Please give more explanations for "they do not transmit simultaneously", given that large-scale MIMO antennas are now relatively common.

4. The author should conduct an analysis in the simulation using a specific video service, ensuring that the parameters of the simulation are explained in a more reasonable manner.

Comments on the Quality of English Language

The English expression of the article is acceptable.

Reviewer 3 Report

Comments and Suggestions for Authors

To start with, the topic is interesting, practical and up to date. ITS, video content delivery and IoT networks are analyzed all around the world. It is surely worth presenting and investigating. However, many information are missing, while some are misleading or omitted. There are many issues that should be properly addressed. In order to raise the overall quality of this paper, Authors are strongly advised to acquaint with the list of suggestions and comments.

Suggestions and comments:

1) To start with, when having all 4 co-authors serving as corresponding authors, consider inserting a statement like [All Authors contributed equally to this paper].

2) To be honest, the Introduction should focus more on the analyzed topic of video transmission, MPEG-4, HEVC, etc. Therefore, do extend it with appropriate citations, especially those related with QoS aspects.

3) Authors are strongly advised to extend the scope and number of cited papers related with video/content consumption, transmission, coding and compression. Do look for both theoretical studies as well as research papers, include user expectations surveys and quality evaluations, etc. Currently, the second part (Related works) is too short, the present form is not acceptable.

4) Section 3 is too short as well, do provide additional info on your method, compare it with existing ones. Justify why is it (or why could it be) better than others.

5) General remark – in its current form, the manuscript lacks proportion. Do look at the length of respective sections and sub-sections – some comprise of only 1 page, while other span across multiple pages. Rethink and reorganize the structure of your paper so that it flows smooth.

6) Do include more info about the utilized dataset – was it only one video sequence? How many files are there included in general? How long did it last? What was it describing? Was it really only resolution? This info should be summarized in a table. Next, Authors could show some frames of the video sequence, so that the reader could have at least basic knowledge (glance of an eye) about the content.

7) What about the utilized laboratory stand during experiments – was it only the OMNET software? What about other simulation environments, tools, or third party libraries? What about the PC hardware components, including: CPU, GPU, RAM, etc.? At least principle technical specs should be provided.

8) How was the PSNR calculated? Was it fully automated by software? A more detailed description about the measurement procedure should be provided.

9) How big was the selected area of the Las Vegas city? How many transmitters, receivers or transceivers did it include? How many nodes/devices were deployed and how were they organized? Did this scenario involve wireless or wired transmission? Then, what was the transmission mode, selected frequency, SNR, BER/BLER, channel characteristics and channel noise, etc.? Did you analyze aspects such as signal de/refraction, scattering, multipath propagation, etc.? What about natural and man-made objects that could influence the signal conditions? What about high buildings and other obstacles? Additional comments are necessary.

10) Was the Las Vegas scenario analyzed only during simulation, or did the Authors perform field-test measurements? What about the traffic caused by pedestrians and motorized people? Additional comments seem necessary.

11) How long did a single session take? How many runs did you check?

12) Why was a US-based scenario chosen, if all Authors come from Spain, and this word was founded by an EU project?

13) General remark – I do understand those plots in the Results section, but I am concerned about the measurement scenario, various set parameters, etc. The current form is doubtful, it needs to be properly justified.

14) The Conclusions part is too short and not convincing at all. Do provide additional feedback on your findings, mention about open issues and future study directions.

15) Taking into account the topic of this paper, the number of cited references is far too short, it needs to be broadened and extended. Do look for additional papers and conference proceedings. The current number and scope is not acceptable.

To sum up, this could be a good paper, yet it requires major revisions and extensions. Authors are therefore strongly requested to prepare a modified version of their initial submission.

Round 2

Reviewer 2 Report

Comments and Suggestions for Authors

Thanks, the authors have answered my concerns.

Comments on the Quality of English Language

The English writing of the paper is good.

Author Response

Dear Reviewer,

We would like to thank you for your valuable advices and constructive suggestions. After addressing all the commented issues, the manuscript has been significantly improved, and we hope that this version could be accepted for publication.

We also want to thank you for the valuable suggestions for future work, which we will take into account.

Thank you for your consideration

Sincerely,

Pablo Garrido

Reviewer 3 Report

Comments and Suggestions for Authors

Authors have prepared a revised version of their initial submission. Now the paper is much more informative and pleasant to read. Being rearranged, it has the right proportions when it comes to respective sections and sub-sections, etc. Changes made are properly highlighted and justified. Indeed, this idea, approach and results are novel, surely worth presenting and publishing in the Journal. I encourage their Authors to continue their studies in raising the QoS/QoE of VANETs, IoTs, as well as other cellular and mobile networks. In the nearest future, be sure to perform field test measurements in various outdoor scenarios and environments. Additionally, do extend the scope and range of cited papers, and do look for other works, both surveys or expectations surveys and simulation or field-test research campaigns.

To sum up, this paper fulfils necessary requirements in order to be accepted and published. Therefore, I do recommend it to be processed further.

Author Response

(The authors gave the same response as above.)
